# Closed-Form Power Normalization Methods for a Satellite MIMO System

**DOI:** 10.3390/s22072586

**Published:** 2022-03-28

**Authors:** Andrea Segneri, Alejandro Baldominos, George Goussetis, Alberto Mengali, Nelson J. G. Fonseca

**Affiliations:** 1Institute of Sensors Signals and Systems, Heriot-Watt University, Edinburgh EH14 4AS, UK; as244@hw.ac.uk (A.S.); ab268@hw.ac.uk (A.B.); 2Telecom Systems and Techniques Section, European Space Agency, 2201 AZ Noordwijk, The Netherlands; alberto.mengali@esa.int; 3Antenna and Sub-Millimetre Waves Section, European Space Agency, 2201 AZ Noordwijk, The Netherlands; nelson.fonseca@esa.int

**Keywords:** multiple-input multiple-output, satellite communications, array-fed reflectors, linear precoding, power normalization techniques

## Abstract

The paper proposes a new set of normalization techniques for precoding/beamforming matrices applicable to broadband multiuser multiple-input multiple-output (MIMO) satellite systems. The proposed techniques adapt known normalization methods to account for the signal attenuation experienced by users due to the degradation of antenna gain and free space losses towards the edge of the coverage. We use, as an example, an array-fed reflector (AFR) antenna onboard a satellite in geosynchronous orbit (GEO), which provides a favorable trade-off between high-directivity, reconfigurability, and the requirement for digital processing, but suffers from high scan losses away from broadside due to optical aberrations when considered for global coverage applications. Three different precoding/beamforming techniques are employed, namely zero forcing (ZF), minimum mean squared error (MMSE), and matched filtering (MF). Low-complexity power normalization techniques digitally applied after the beamformer are introduced that, in the absence of any atmospheric effects, lead to iso-flux-like characteristics whilst satisfying the power constraint per feed. In comparison with other methods reported in the literature, mainly based on iterative algorithms, the proposed techniques consist in closed-form expressions to provide uniform signal-to-noise ratio (SNR) and signal-to-noise plus interference ratio (SNIR) across the users without significant impact on the payload sum rate. Numerical results are presented to comparatively demonstrate the achieved performance in terms of total capacity and distribution of SNR and SNIR at various noise and interference scenarios.

## 1. Introduction

The objective of increasing the throughput at a competitive price in broadband satellite communication systems has triggered investigations of terrestrial solutions in the satellite context. Based on the results of Dirty Paper Coding (DPC) [1], and the vast exploitation of multiuser multiple-input multiple-output (MIMO) terrestrial systems, a promising technique consists in adopting precoding to cancel the interference, allowing more aggressive frequency reuse schemes for the satellite forward link. The extended version of the digital video broadcasting (DVB-S2X) standard, with its novel superframe structure, supports the implementation of satellite-based precoding [2]. Research has addressed the application of this technique in single feed per beam (SFPB) antennas [3,4,5]. Many drawbacks of such systems have been highlighted. Due to a limited payload processing capability, precoding has to be implemented at the gateways (GWs); if the GWs are not interconnected, inter-cluster interference arises since each GW can only process a subset of beams, limiting the benefits of precoding [6]. Precoding relies on the knowledge of the channel matrix; typically, users have to estimate and report the channel state information (CSI) via a return link, which, however, can contain errors and be outdated. Another issue is the non-linearity introduced by the on-board high power amplifiers (HPAs) [7]; typically, HPAs need to be operated as close as possible to the amplifier compression point to optimize power efficiency [8]. The application of precoding may change the power assigned to each antenna feed depending on the channel characteristics, making the back-off requirement harder to satisfy. Recently, the concept of massive MIMO employing active antennas is also being analyzed for applicability in satellite communications [9,10] and, in general, in non-terrestrial networks (NTNs) [11]. The difficulties that have been summarized are even more challenging for massive MIMO systems; for example, the CSI should be estimated at the receiver and signaled back to the GW individually for each feed of the large-scale antenna array. In [12], a pragmatic approach for exploiting massive MIMO with much lower complexity than precoding has been introduced; CSI is estimated by modeling the antenna pattern and the downlink channel for a fixed set of beams. The precoding/beamforming coefficients are computed based on these fixed pointing directions. By properly scheduling the users to be served in a time division multiplexing (TDM) scheme, it is demonstrated that such a pragmatic solution can achieve a performance close to traditional precoding techniques. In a similar manner, in this paper, we are interested in pragmatic applications of precoding to satellite MIMO systems and we consider ideal, previously estimated CSI characteristics to assess the proposed power normalization techniques in a reference scenario employing open-loop precoding strategies.

The active antenna considered in this paper is based on an array fed reflector (AFR) configuration with distributed amplification. Such antenna architecture provides high directivity and flexibility at a moderate complexity and cost [13], and is the preferred solution for next-generation software-defined satellites in geosynchronous orbit (GEO). Despite these advantages, the optics inevitably lead to higher scan losses [14,15]. This is particularly evident in the hybrid optics described in [16,17], with higher scan losses in the imaging plane compared to the focusing plane. The directivity degradation at increased pointing angles is predominantly due to spill-over losses and optical aberrations. Possible methods to mitigate this issue are based on shaping the AFR geometry to obtain an isoflux-like performance, and are described in patent [18]. System performances are driven by the power flux density achieved across the service area, which is a combination of antenna gain and payload power distribution. Thus, the correction proposed at the antenna level in [18] could equally be considered at the payload level.

In this paper, we explore the possibility of recovering scan losses at the payload level by computing suitable precoding/beamforming weights that will assign more power to users located in beams with higher depointing angles, thereby equalizing the received signal-to-noise ratio (SNR). Combined with a user scheduling protocol, the proposed techniques consistently reduce signal-to-noise plus interference (SNIR) variability, hence providing a similar quality of service among users. The proposed methods also consider free space losses (FSL), that, like scan losses, are greater towards the edge of coverage. Zero forcing (ZF), minimum mean squared error (MMSE), and matched filtering (MF) are considered for obtaining the precoding/beamforming matrix prior to normalization. It has to be highlighted that ZF and MMSE performance are not achievable in practice due to the aforementioned drawbacks; however, these precoding techniques are common in MIMO systems and they are evaluated to provide reference upper bounds. MF can be adopted in actual satellite MIMO systems following the pragmatic approach in [12].

In [19], a power normalization after precoding, referred to as CTTC, is introduced to satisfy the power per feed constraint and to exploit all available power at the cost of some co-channel interference. Whilst the performance in terms of sum rate is favorable, with this approach, the SNIR towards the edge of coverage is consistently compromised. Here, we propose three pragmatic normalization techniques that maintain a uniform power per feed whilst providing similar sum rates and, together, a more fair SNR and SNIR distribution across the full earth coverage. A first approach, in the remaining referred to as Loss Mitigation, makes use of the known antenna characteristics, in this case the designed AFR, to manipulate the precoding matrix. The second approach, SNR Equalization, targets an equal received power per user. Similarly, the last approach, Strict SNR Equalization, relaxes the power per feed constraint to provide exactly the same SNR per user; in this case, the constraint per feed is not satisfied. However, we will show in Section 4 that the variability of the power per feed is drastically reduced. Typically, the joint power per feed and equal SNIR constraint is treated introducing an optimization problem that requires iterative algorithms to be solved [20]. In [21], the general non-convex optimization problem to maximize the minimum SNIR, max-min SNIR, under equal power per feed constraint is expressed and is reformulated as a convex one that can be solved via iterative algorithms. In [22,23], optimization problems targeting desired performance are investigated under a sum power constraint. We demonstrate that by employing the proposed closed-form techniques, with proper user scheduling, fairness can be increased without significant additional complexity. In [24], pragmatic solutions to power normalization problems are analyzed for various precoding techniques. However, the effect of the user scheduling on SNIR variability has not been treated and the system assumptions refer to terrestrial MIMO applications.

The impact of the normalization techniques on system performance, combined with the different precoding schemes, are analyzed in terms of received SNR and SNIR per user; it is shown that by using Sum Power and CTTC, the performance is highly correlated with the geographical location of users and can be drastically equalized by adopting the proposed methods. At different noise levels, the total throughput, SNR, and SNIR variability among users are evaluated using a Monte Carlo approach. It is shown that the iso-flux-like characteristic and a more uniform SNIR among users are obtained in all the analyzed scenarios, while the total throughput is not compromised. The main contributions of this work can be summarized as follows:Application of precoding to AFR antennas in a broadband satellite MIMO system;Introduction of three novel power normalization methods accounting for the signal attenuation towards the edge of the coverage due to the considered satellite communication characteristics;Demonstration of reduced SNR and SNIR variability among users when combining user scheduling with the proposed power normalization methods;Analysis of different power normalization methods applied to various linear precoding/beamforming techniques (ZF, MMSE and MF).

The paper is organized as follows. In Section 2, the system model, user scheduling, and precoding techniques are detailed, together with system assumptions. The normalization methods are described in Section 3. In Section 4, the simulation results are presented, and in Section 5, the conclusions are drawn.

## 2. System Model

In order to benchmark the proposed methodologies based on precoding matrix normalization, we consider the forward link of a broadband satellite system operating in GEO. The satellite is equipped with an AFR having distributed amplification (here, assumed to be one amplifier per feed) and an On-Board Processor (OBP) to drive the beamforming network and produce the multibeam coverage. The focus is on the downlink path of the system. TDM is considered, such that at each time epoch, a subset of users is served, thus adopting a full frequency reuse (FFR) scheme. We assume, for simplicity, a number of users equal to *K* and one user per beam. Furthermore, all users are equipped with a single antenna; the transmitting AFR instead possesses *N* feed elements, which are used to produce the beams. The antenna geometry is an imaging configuration in which all feeds contribute to each beam. The communication system can be modeled by introducing a MIMO channel model. Denoting x as the K×1 unit energy signal vector intended to users and U as the N×K precoding/beamforming matrix, the N×1 vector containing the complex signal transmitted by the *N* radiating elements is
(1)$$y=\sqrt{P}Ux,$$
where *P* is the total payload RF power. The received signal vector of *K* elements is then
(2)$$z=Hy+n=\sqrt{P}\mathrm{HUx}+n,$$
with H being the K×N channel matrix and n being the K×1 Gaussian noise vector representing receiver noise. We assume, without loss of generality, that each user experiences the same noise power.

### 2.1. Channel Model

The channel matrix, representing the overall complex transfer function, can be evaluated by taking into account the satellite propagation characteristics. The downlink channel operating under line-of-sight (LOS) is modeled by characterizing the transfer function of each feed element to the desired directions and by including the propagation fading. FSL represents an important fading effect that, like the scan loss, adds another source of gain imbalance, reaching the maximum for users that are located near the edge of coverage. We neglect other sources of perturbation, such as rain fading, and focus on the stationary condition as assumed in [12].

The characterization of the total field received by the *K* users depends on the AFR configuration and is detailed as follows. The AFR considered in this paper has been optimized to provide full earth coverage from GEO and to possess the following characteristics: no feed blockage, reduced size, maximization of directivity, reduced scan loss, and grating lobes. The reflector is illuminated with an array of 511 circularly polarized feeds placed in a hexagonal lattice. The antenna geometry is depicted in Figure 1 and the design parameters are reported in Table 1.

This antenna design results in a 3 dB beamwidth of 0.8 degrees, with a peak directivity of 46.5 dB and a maximum scan loss of 2.4 dB.

An in-house tool developed in Heriot-Watt University is used to characterize the AFR antenna [25]. This tool uses Physical Optics (PO) to obtain the far field of every feed in the required directions and implements acceleration methods to reduce the computational effort [25,26]. In this paper, the far field of every feed is computed for a fixed set of points. Next, interpolation is performed to estimate the complex copolar component in every user direction. This method accelerates the estimation of the channel matrix, which is required in every simulation.

Let *K* be the number of users to be simultaneously served in a time slot, represented as *K* points in a [u,v] satellite coordinate system. The corresponding component of the far field for the point $\left(u_{k},v_{k}\right)$ is computed for each of the *N* feeds. These computed values can be disposed to form a K×N matrix $E^{far}=E_{n}^{far}\left(\theta_{k},\phi_{k}\right)$, with 1≤k≤K and 1≤n≤N, where [12]:(3)$$\theta_{k}=\sqrt{u_{k}^{2}+v_{k}^{2}},\;\phi_{k}=\tan^{-1}\left(\frac{v_{k}}{u_{k}}\right).$$

$E^{far}$ represents the transfer function from antenna feeds to users in the far field without any channel fading. In order to include the FSL, let $l^{fs}$ be the vector with *K* elements, representing the FSL for each user, computed as [27]:(4)$$l_{k}^{fs}={\left(\frac{\lambda }{4\pi d_{k}}\right)}^{2},$$
with λ being the wavelength and $d_{k}$ being the distance satellite-user. The overall channel matrix from feeds to user can then be modeled as
(5)$$H=\mathrm{diag}\left\{l^{fs\frac{1}{2}}\right\}E^{far},$$
where the diag{·} operator applied to a vector returns a square matrix having the vector elements in the diagonal, and the square root of $l^{fs}$ indicates the element-wise operation. The set of *K* users, defining the user distribution and, hence, the channel matrix that is used in each Monte Carlo iteration is based on the results of [12], where it is found that separating the users by a minimum distance, the Poisson disk radius, brings substantial benefits in terms of system performance. While the Poisson disk distribution is impossible to achieve in practice, a similar performance can be achieved with appropriate RRM techniques as shown in [28]. In this paper, to avoid the need of performing an RRM optimization at each simulation, we generated an approximate Poisson disk distribution by sampling a larger set of points uniformly distributed in the region of interest (ROI), corresponding to the satellite coverage region. The algorithm to perform the sampling that ensures a minimum distance between users is detailed in [29].

### 2.2. Precoding

To compute the precoding/beamforming matrix from the modeled channel matrix, we focus on linear techniques that, even if sub-optimal, provide significant capacity improvement without requiring the processing complexity of non-linear techniques [30] and constitute a practical choice for satellite systems [31]. In the following, three linear precoding methods are described; the resulting N×K matrix is denoted W and the techniques are identified by related subscript. The following precoding methods do not account for payload power limitations, so a further step is required to obtain the precoding matrix U of Equation (Equation 1).

ZF precoding is an effective way of canceling the interference between users. By inverting the channel, the received signal is forced to be as close as possible to the desired transmitted signal x. The effect is that nulls are placed in the interference directions. If the channel matrix is full rank, the ZF precoding weights can be obtained by [32]:(6)$$W_{ZF}=H^{H}{\left(HH^{H}\right)}^{-1},$$
where $H^{H}$ denotes the Hermitian transpose of H. In general, it can be derived as the Moore–Penrose pseudo inverse $W_{ZF}=H^{+}$. ZF precoding allows the users to recover their intended signals without interference from other beams. However, in [33], it was proved that ZF can cause a major degradation of system performance, especially in scenarios with a high number of users and noise. Another practical choice is to relax the condition of having zero interference for all the receivers by regularizing the inverse, adding a scaled identity matrix before inverting. The regularized inversion was introduced in [33] and can also be obtained from Minimum Mean Square Error (MMSE) optimization problems [34,35]. The MMSE precoder is computed as
(7)$$W_{MMSE}=H^{H}{\left(HH^{H}+\alpha I\right)}^{-1},$$
where α≥0 is the regularization factor. An optimization problem can be formulated based on various criteria; in [33], the optimal factor to maximize the SNIR at the receivers was derived as $\alpha =K\sigma^{2}/P$, where $\sigma^{2}$ is the noise variance.

The last precoding/beamforming technique considered is MF; this beamforming technique maximizes the gain of the array towards the users, and can be interpreted as steering the beams in the user directions [36]. However, it does not take into account any interference mitigation. The beamforming matrix is computed as [37]:(8)$$W_{MF}=H^{H}.$$

Even if it does not take into account interference or noise, it is a very low complexity practical choice and can represent the basis for a pragmatic implementation of Massive MIMO in the satellite context [12].

The precoding/beamforming matrix needs to be normalized to account for payload limitations, such as total available RF power. Another important aspect is the power variation across the feed array that can be very large when linear precoding is used, resulting in some HPAs operating at high backoff. This reduces the efficiency of the DC to RF power conversion and constitutes an important draw-back of precoding application [7]. In the following, we focus on power normalization criteria that assign uniform power among antenna feeds at the expense of some co-channel interference. In [19], a matrix normalization satisfying this requirement, named CTTC, was introduced and in [12], it was shown to provide excellent performance compared to other normalization techniques in various scenarios. In this paper, we also include Sum Power normalization as a benchmark, corresponding to the simple normalization of the precoding matrix to satisfy the sum power constraint. The normalized precoding/beamforming matrix will be denoted by U, in accordance with Equation (Equation 1).

Given the total satellite power, P, the effective power allocated to users and antenna feeds can be derived from U and represented as vectors of *K* and *N* elements, respectively, as
(9)$$p^{user}=\left[p_{1}^{user},p_{2}^{user},\ldots ,p_{K}^{user}\right],\;p^{feed}=\left[p_{1}^{feed},p_{2}^{feed},\ldots ,p_{N}^{feed}\right],$$
where
(10)$$p_{k}^{user}=P∥u_{k}^{C}{∥}^{2}=P\sum_{i=1}^{N}{\left|u_{ik}\right|}^{2},$$
(11)$$p_{n}^{feed}=P∥u_{n}^{R}{∥}^{2}=P\sum_{j=1}^{K}{\left|u_{nj}\right|}^{2},$$
for 1≤k≤K and 1≤n≤N. $u_{k}^{C}$ represents the *k*-th column vector of matrix U, $u_{n}^{R}$ the *n*-th row, and ∥·∥ the Euclidean norm operator. All normalization methods require that
(12)$$\sum_{k=1}^{K}p_{k}^{user}=\sum_{n=1}^{N}p_{n}^{feed}\leq P,$$
so the total power constraint is satisfied. The power per feed constraint has the simple form $p_{n}^{feed}\leq P_{n}$, for each *n*, where $P_{n}$ is the power constraint on the *n*-th feed; we consider the case where each feed has the same constraint, hence $P_{n}=P/N$. The normalization can follow different strategies that are detailed in the next section. Given a normalized matrix U, we can express the formulation of signal-to-noise, interference-to-noise, and signal-to-noise plus interference ratio experienced by user *k* as [4,12]
(13)$$SNR_{k}=\frac{P|h_{k}^{R}u_{k}^{C}{|}^{2}}{N_{0}B_{w}},$$
(14)$$INR_{k}=\frac{P\sum_{j=1,j\neq k}^{K}{\left|h_{k}^{R}u_{j}^{C}\right|}^{2}}{N_{0}B_{w}},$$
(15)$$SNIR_{k}=\frac{SNR_{k}}{1+INR_{k}},$$
where $N_{0}$ is the noise power density and $B_{w}$ the total bandwidth. Once the $SNIR_{k}$ for each user is obtained, the throughput is derived from the spectral efficiency table of the DVB-S2X standard [31]; hence, the total throughput is
(16)$$Th=B_{w}\sum_{k=1}^{K}\eta_{DVBS2X}\left(SNIR_{k}\right).$$

In Section 4, the performance of the proposed methods will be evaluated in various interference and noise scenarios. In order to demonstrate the reduced SNR and SNIR variability, obtained by the precoding and matrix normalization methods in the simulated scenarios, we will present their Cumulative Density Function (CDF) and dynamic ranges, i.e., $\max\left(SNR_{k}\right)-\min\left(SNR_{k}\right)$ and $\max\left(SNIR_{k}\right)-\min\left(SNIR_{k}\right)$, respectively. The steps to produce these results are summarized in Algorithm 1. The number of feeds (N=511) is fixed by the chosen antenna configuration, as well as the parameters reported in Table 1.
**Algorithm 1** Iterative evaluation of performance results.**Input:** 
*K*, *P*, $N_{0}$, $B_{w}$ 1:**for all** Monte Carlo iterations **do** 2:    Generate a large set (10×K) of uniformly distributed [u,v] points in the ROI. 3:    Sample the uniform set to obtain the approximated Poisson distribution of *K* points, as described in [29]. 4:    Compute the far field related to feed *n* and point $\left(u_{k},v_{k}\right)$ for all k=1,…,K and n=1,…,N, as described in [26]. 5:    Compute the free space loss using Equation (Equation 4) for all k=1,…,K. 6:    Obtain the channel matrix H, Equation (Equation 5). 7:    Compute precoding matrices $W_{ZF}$, $W_{MMSE}$ and $W_{MF}$, Equations (Equation 6)–(Equation 8). 8:    Compute normalized precoding matrices $U_{ZF}$, $U_{MMSE}$ and $U_{MF}$ for each normalization method detailed in Section 3. 9:    Compute $p_{k}^{user}$, $p_{k}^{feed}$, $SNR_{K}$, $SNIR_{K}$ and Th for all precoding/normalization combinations, Equations (Equation 10), (Equation 11), (Equation 13), (Equation 15) and (Equation 16). 10:**end for** 11:Compute the average of the metrics over the number of Monte Carlo iterations.**Output:** 
average $p_{k}^{user}$, $p_{k}^{feed}$, $SNR_{k}$, $SNIR_{k}$ and Th for all precoding/normalization combinations.

## 3. Power Normalization Methods

In this section, five matrix normalization methods are presented, each of them emphasizing some conditions. The first one, Sum Power, typically achieves the best performance in terms of throughput and will be used as a reference. The remaining methods aim at:Uniform power per antenna feed;Total throughput not severely penalized w.r.t Sum Power;Reduced SNR and SNIR variation among users;All available RF power exploited;Applicability to all the discussed precoding/beamforming methods;Closed-form expression, low-complexity technique.

The normalization methods are divided into sub-steps; equations are presented per row or column for notation simplicity and must be performed for all rows and columns, i.e., for each 1≤k≤K and 1≤n≤N.

### 3.1. Sum Power

The first normalization considered is Sum Power. The complex matrix W is simply scaled to satisfy the equality of the sum power constraint in Equation (Equation 12):(17)U=γW,
where $\gamma =1/\sqrt{\mathrm{trace}(WW^{H})}$. Note that the properties of the precoding matrix W are not changed since we are simply scaling by a constant value. When ZF is applied as the precoding technique, for N>K cases, the channel inversion forces the $SNR_{k}$ and $SNIR_{k}$ to be identical for all users, while the power per user and per feed, vectors $p^{user}$ and $p^{feed}$, exhibit a great variation. The $SNR_{k}$ and $SNIR_{k}$ uniformity is broken by adopting MMSE, which introduces a regularization factor in the inversion, or by applying MF, where no interference management takes place.

### 3.2. CTTC

CTTC is a normalization proposed in [19], as a variation of the Taricco method in [3], which ensures that users get the same RF power while the power per feed constraint is not violated; this implies that the total power is generally lower than the available RF power while the interference is not affected. The CTTC, instead, tries to exploit the unused power by normalizing the antenna feed power so that all array elements get the same RF power at the expense of some co-channel interference mitigation. The first step consists of normalizing the power per user [38]:(18)$${\tilde{u}}_{k}^{C}=\frac{w_{k}^{C}}{\sqrt{K}∥w_{k}^{C}∥},$$
and the second step corresponds to a normalization of power per feed, i.e., all the row vectors of the precoding matrix with
(19)$$u_{n}^{R}=\frac{{\tilde{u}}_{n}^{R}}{\sqrt{N}∥{\tilde{u}}_{n}^{R}∥}.$$

Note that scaling with factors $1/\sqrt{K}$ and $1/\sqrt{N}$ ensures that the constraint is satisfied after each step. Performing the feed normalization at the last step ensures a uniform power distribution among feeds, while the power per user can vary substantially, as well as the SNR and SNIR, as will be shown in Section 4. This applies to all the precoding/beamforming techniques previously discussed. The following three methods propose a variation of the CTTC, aiming at simultaneously having uniform power per feed and uniform SNR per user. The CTTC steps are reproduced in matrix form in Table 2.

### 3.3. Loss Mitigation

Users typically experience different amounts of losses that depend on their positions; for example, the ones that are closer to the edge of the coverage suffer up to 2.4 dB of scan losses due to the considered antenna characteristics. This is compensated by using ZF and Sum Power as previously observed; however, the uniform power per feed is not satisfied and other precoding techniques do not recover the losses. Other precoding approaches, for example based on max-min SNIR optimization [22], are possible; however, the complexity level would be higher. The idea of this matrix normalization is to modify the first step of CTTC in order to assign more power to users experiencing more losses due to their locations. The first step is realized by a normalization of power per user weighted following the relation with user radial distance losses, plotted in Figure 2,
(20)$${\tilde{u}}_{k}^{C}=\frac{f\left(\rho_{k}\right)w_{k}^{C}}{\sqrt{K}∥w_{k}^{C}∥},$$
with $\rho_{k}=\sqrt{az_{k}^{2}+el_{k}^{2}}$ representing the radial distance of user *k*, $az_{k}$ and $el_{k}$ being the azimuth and elevation angle; f(·) is the function representing the scan losses plus the propagation losses, as in Figure 2, in natural values. Note that f(·) has a slight asymmetry in az due to the offset geometry of the considered antenna, but that can be ignored since it results in negligible impact on performance. In case of higher asymmetry, the method can be generalized characterizing the losses as a function of azimuth and elevation, f(az,el). The second step is exactly the same as the second step of CTTC in Equation (Equation 19). Table 3 reports the steps.

### 3.4. SNR Equalization

The introduced Loss Mitigation can mitigate the losses which are simply estimated knowing the geographic position of a user. However, it does not take into account the user distribution and associated impact on the SNR. Heuristically, users that are very close together will experience higher interference and, at the same time, lower SNR, since some power is wasted in the interference direction when compared to precoded signal transmission adopting ZF and MMSE. The SNR Equalization method also compensates against these gain losses. It is noted that if two users are too close, a large amount of power will be used to compensate their gains at the expense of the remaining users, lowering the performance of the overall system. A user scheduling that ensures a homogeneous distance between users, such as the Poisson distribution, is thus required. While CTTC imposes equal transmitted power per user at the first step, this normalization method forces the received power per user to be identical. The first step imposes an equal SNR, (see Equation (Equation 13)), by
(21)$${\tilde{u}}_{k}^{C}=\frac{w_{k}^{C}}{s_{k}},$$
with $s_{k}=\left|h_{k}^{R}w_{k}^{C}\right|$, the signal amplitude received by user *k*. The second step is identical to the second step of CTTC, Equation (Equation 19). Table 4 indicates the steps.

### 3.5. Strict SNR Equalization

Although the previous method will reduce the SNR variation across the users, the normalization of all row vectors to $1/\sqrt{N}$ performed in the second step (to ensure that all the array feeds operate at the same power level) renders Equation (Equation 21) approximate. With the Strict SNR Equalization technique, we first impose equal power per feed and then equal SNR per user. In this way, equal SNR across the users is achieved at the expense of some variation in the power per feed. This technique is applied in three steps. The first step imposes uniform power per feed:(22)$${\tilde{u}}_{n}^{R}=\frac{w_{n}^{R}}{\sqrt{N}∥w_{n}^{R}∥},$$
the second step realizes the equal SNR per user:(23)$${\bar{u}}_{k}^{C}=\frac{{\tilde{u}}_{k}^{C}}{{\tilde{s}}_{k}},$$
where ${\tilde{s}}_{k}=\left|h_{k}^{R}{\tilde{u}}_{k}^{C}\right|$. The last step rescales the modified beamforming matrix to have the total power equal to *P*, without further modifying matrix properties:(24)$$U=\frac{\bar{U}}{\sqrt{\mathrm{trace}(\bar{U}{\bar{U}}^{H})}}.$$

In Table 5, all the steps are reproduced.

## 4. Results

In order to evaluate the performance of the normalization techniques discussed above, Monte Carlo simulations were performed by varying the users distribution and the results are presented in this section. We show the performance of the proposed precoding/normalization techniques in typical satellite noise scenarios. The system parameters considered are reported in Table 6.

The number of Monte Carlo iterations is set to five, since it is found to be an adequate number of simulations for obtaining reliable results of such systems. In [28], it is claimed that a good throughput accuracy, averaged on the number of Monte Carlo trials, is already achieved with five iterations; a similar convergence was experimentally confirmed with our system assumptions. As an example, the resulting throughput with 255 users adopting MF and Loss Mitigation with only one iteration is 138.775 Gbps, with five iterations is 138.975, and with 10 iterations is again 138.975, showing a very limited impact on performance results when varying the number of trials.

We have selected two scenarios for presenting the results. A scenario with a limited number of users (N/K≈2), where the noise is the predominant factor, and one with more users and, thus, more interference (N/K≈3/4). We first show the reduced SNR and SNIR variability among users and then ensure that the total throughput is not compromised in various noise level scenarios.

### 4.1. Case N/K≈2

The number of users K=255 has been fixed, corresponding to a ratio for the number of array feeds versus the number of users of N/K≈2. These users are sampled from a larger set of uniformly distributed users as explained in Section 2. We first present the effect of the normalization techniques on the SNR and SNIR per user for all the discussed precoding/beamforming methods. In a first plot, the obtained SNR, or SNIR, per user is depicted in relation to the user position in the ROI (in azimuth and elevation angles), while in a second plot, we show the resulting CDF for SNR and SNIR in the related section. We also assess the techniques at different noise scenarios, controlling the receiver noise. For the selected receiver noise level, the $SNR_{k}$ and $SNIR_{k}$ are computed for each user, *k*, and the average SNR is thus $SNR_{avg}=\sum_{k}SNR_{k}/K$. For the system parameters of Table 6, the reference SNR is around 5 dB. We will consider $SNR_{avg}$ values in the range [−2, 9] dB to test the proposed techniques in typical noise scenarios that can be experienced if different parameters are assumed, such as lower payload power or different G/T values. Tables will show the resulting maximum and minimum dynamic range of SNR and SNIR obtained by the Monte Carlo simulations over the different noise levels to ensure that the iso-flux characteristic and reduced SNIR variability remain valid.

### 4.2. SNR Results

Figure 3 shows how the SNR per user is affected applying the proposed methods. As previously noted, the Sum Power normalization technique applied to ZF leads to a uniform SNR, which is confirmed by the first plot on the top-left corner. However, when applying CTTC to ZF, the SNR variation is highly increased (second plot on the top); the precoding/beamforming properties of ZF are broken by the CTTC normalization steps and this leads to enhanced performance, w.r.t Sum Power, for users near the center of the coverage at the expense of users towards the edge. This is caused by the scan and propagation losses that Sum Power recovers by assigning more power to the users with higher losses. As can be appreciated from Figure 3, the proposed normalization methods deliver a uniform service to the users, independently of their location. The resulting SNR variation among users is highly reduced and it is almost nulled, except for the ZF-Loss Mitigation combination. This is due to the fact that ZF manages very close users by completely suppressing reciprocal interference, a condition that is broken by the normalization steps.

Figure 4 confirms the enhanced SNR uniformity by presenting the CDF of the SNR per user.

Table 7 summarizes the SNR dynamic ranges for the considered simulated noise scenarios, related to $SNR_{avg}$ values from −2 dB to 9 dB.

The minimum and maximum of the dynamic ranges are taken over the noise levels to synthetically show the dependence of SNR uniformity on signal-to-noise scenarios. As expected, the SNR ranges are affected by noise levels only when considering the MMSE precoding scheme, since SNR determines the regularization factor when computing the MMSE matrix in Equation (Equation 7); however, the normalization techniques that impose an equal power per feed (all except Sum Power) make this dependence on SNR ranges less important. The difference between maximum and minimum SNR values when considering the CTTC is around 3.5 dB. That is exactly the scan and free space loss of the considered satellite AFR system at the edge of coverage (Figure 2); these losses can be consistently mitigated adopting the proposed normalization techniques.

#### 4.2.1. SNIR Results

In Figure 5 and Figure 6, we show that the reduced SNR variation also implicates a more uniform SNIR per user in the considered scenario. Although the SNIR per user is not constant as in the SNR case, the dynamic range of SNIR is also significantly reduced. This is because the interference is not playing an important role, thanks to the Poisson distribution of users obtained with adequate user scheduling. It has to be remarked that a Poisson distribution of users obtained by a large set of uniform distributed users is a favorable condition that cannot be obtained in practice. Nevertheless, a scheduling method to approach the general Mixed Integer Quadratic Programming (MIQP) optimization problem with affordable complexity is presented in [28], thus optimally selecting the users per slice, limiting their reciprocal interference. In Section 4.3, we examine the case with more users, and, thus, more interference.

Figure 5 presents the SNIR distributions based on the geographical location of users. As shown for the SNR results, ZF precoding combined to Sum Power normalization ensures a constant SNIR per user. This is invalidated by the CTTC normalization method, which causes an unbalanced service to users for all the precoding/beamforming techniques, due to the favorable signal propagation characteristics near the center of the coverage. Instead, by adopting the proposed normalization methods, the SNIR variability is greatly reduced. This can be quantitatively deduced by looking at the CDF of the estimated SNIR per user in Figure 6: the SNIR dynamic range is reduced by almost 3 dB with respect to CTTC when using ZF, MMSE, or MF schemes for the investigated noise scenario. Moreover, the three proposed techniques perform similarly when applying MMSE and MF schemes, while there are some differences in the ZF case for Loss Mitigation with respect to SNR Equalization and Strict SNR Equalization. This is due to the aforementioned ZF effect of exacting nulling interference.

In Table 8, the dynamic ranges of SNIR for different noise scenarios are reported. Obviously the dependence on the SNIR ranges on noise levels is stronger than SNR-related results (Table 7); the advantages of the proposed techniques on SNIR variability are the greatest for the lowest signal-to-noise levels, where $SNR_{avg}$ is around −2 dB, which is where interference plays a minor role. Moreover, it is noted that the precoding scheme that tries to completely suppress interference (ZF) results in less SNIR variability over all the considered noise levels, while MF, combined with the proposed techniques, results in noise scenarios ($SNR_{avg}$ around 9 dB) with the highest SNIR dynamic ranges, but still lower than CTTC maximum range.

#### 4.2.2. Throughput Results

In this section, we analyze the performance in terms of throughput, which is plotted versus the average SNR experienced by the users, the same $SNR_{avg}$ values considered in Table 7 and Table 8. The total throughput is then derived from Equation (Equation 16). In Figure 7, it is shown that all the normalization methods have a similar performance in terms of total throughput for most of the different noise levels considered. There are a few noise scenarios where the proposed normalization techniques perform slightly worse than CTTC (and some slightly better), but the capacity reduction is limited to 7% in the worst case, while in the majority of the $SNR_{avg}$ values, the throughput results of the proposed techniques compared to CTTC are practically the same. While the total throughput is not compromised, the SNR and SNIR variability experienced by users is reduced, especially at low $SNR_{avg}$ values. This causes an enhanced throughput for users that are located towards the edge of the coverage, at the expense of users around the center, resulting in a balanced service over the complete coverage.

#### 4.2.3. Power per Feed Dynamic Range in Strict SNR Equalization

The CTTC, Loss Mitigation, and SNR Equalization techniques satisfy the constraint on the power per feed, since the last step in these techniques consists of setting the power per feed equal to P/N. On the other hand, in Sum Power, we consider a sum power constraint and in Strict SNR Equalization, the constraint per feed is relaxed. We report in Table 9 the differences in dB of the power per array feed compared to the uniform power for all the simulated scenarios of Figure 7.

Clearly, CTTC, Loss Mitigation, and SNR Equalization have zero values, since there is no variation of the power per feed. With Strict SNR Equalization, the variation is consistently reduced with respect to Sum Power, while forcing exactly the same SNR among users. However, results in terms of SNR and SNIR variability are approximately the same compared to SNR Equalization. As it can be noted from the CDFs in Figure 4 and Figure 6, there is some slight difference regarding the SNR distribution, while the SNIR distribution obtained with Strict SNR Equalization exactly matches the one obtained with SNR Equalization for MMSE and MF. Since SNR Equalization always ensures the constant power per feed, it is certainly preferable compared to the relaxed constraint per feed version, but it was considered as a reference method capable of exactly imposing equal SNR per user.

### 4.3. Case N/K≈3/4

In this section, we consider a scenario with more users in the ROI, and, thus, more interference. Figure 8 shows the total throughput results with 380 users to be served.

As expected, ZF performance in this scenario is highly compromised, indicating that ZF is not applicable and it will not be discussed. The improvements on the reduction of SNIR variability when applying the proposed techniques are smaller for MMSE and MF beamforming compared to the case with 255 users; however, as it can be seen from Table 10, the scan and free space losses are correctly mitigated, providing the isoflux-like characteristics, especially when applying the MF beamforming scheme. In Table 11, it can be seen that the augmented interference makes the effect on the equalization of SNIR performance limited, but the reduction of the SNIR unbalance is also confirmed in this unfavorable interference scenario, especially at low signal-to-noise levels. However, the similar throughput obtained with the proposed normalization techniques is still valid in this scenario; it should also be noted that the average distance between users in this case is around 0.55 degrees, which is smaller than the 3 dB beamwidth provided by the considered AFR antenna (0.8 deg), and it does not represent a practical case but an adverse scenario to validate the normalization methods in a high interference situation.

## 5. Discussion

Low complexity normalization techniques that enhance performance fairness, thus reducing SNIR variability by providing an iso-flux-like characteristic, have been introduced. Simulations show that the techniques can be applied to ZF, MMSE, and MF precoding/beamforming methods, and can successfully equalize scan and free space losses induced by the reflector antenna and propagation characteristics.

Combined with the proposed normalization techniques, the performance of ZF and MMSE precoding, given their wide application in MIMO systems, have been assessed to provide reference capabilities; however, such precoding schemes are considered unpractical solutions for satellite systems based on active antennas. In fact, system complexity is highly increased under various aspects and performance advantages are limited, as demonstrated in [12]; in the same paper, a pragmatic approach based on MF, namely fixed multi-beam (MB), is presented to exploit satellite MIMO systems. The proposed normalization techniques can also be applied to the MB approach, including in satellite communication systems based on Direct Radiating Array (DRA) that can experience non-negligible scan losses [39]. More broadly, the proposed techniques are applicable to multibeam communication systems relying on line of sight links and typically in mm-wave frequencies. In principle, the methods also apply for non-geostationary orbit (NGSO); nevertheless, the more dynamic channel conditions make the implementation of precoding even more challenging, and pragmatic approaches that do not require channel state information are investigated [10].

The SNR Equalization and Strict SNR Equalization add another matrix multiplication in the normalization process since the channel matrix is considered to equalize the received power, while the Loss Mitigation method is only based on fixed antenna and propagation characteristics, and, thus, adds no extra processing complexity. However, all three methods, being based on closed-form expressions, are affordable in combination with the mentioned pragmatic MB approach, where fixed sets of beams are considered.

The presented results confirm the equalized signal strength performance experienced by users all over the coverage. In particular, Loss Mitigation adopted after MF provides a good trade-off between performance and complexity: the SNR dynamic range is reduced by more than 3 dB compared to CTTC, and almost 7 dB with respect to ZF, for all noise scenarios (Table 7). The SNIR distribution obtained with MF-Loss Mitigation also matches the performance when applying SNR Equalization and Strict SNR Equalization for the considered satellite system, resulting in a reference SNR around 5 dB (Figure 6) with a reduction of the SNIR dynamic range around 3 dB. The benefits of ZF and MMSE on SNIR variability are only visible for higher signal-to-noise levels, in more interference-limited scenarios evaluated (Table 8).

The analyzed effect on SNIR variability of the proposed normalization techniques in various interference and noise scenarios greatly depends on user locations. As previously noted, a Poisson distribution can be a favorable scenario since it maximizes the minimum distance between users; a more realistic user distribution, while approaching the optimal RRM solution with affordable complexity, can be obtained by applying the heuristic RRM (H-RRM) presented in [28]. Moreover, non-uniform traffic conditions should be assessed. The analysis of the proposed techniques, providing the iso-flux characteristic, on such scenarios implementing the pragmatic MB approach and the H-RRM constitutes an interesting research direction. Another idea for future work is to combine the proposed concept with more advanced antenna systems, including for instance reflector shaping [18].

## Figures and Tables

**Figure 1 sensors-22-02586-f001:**
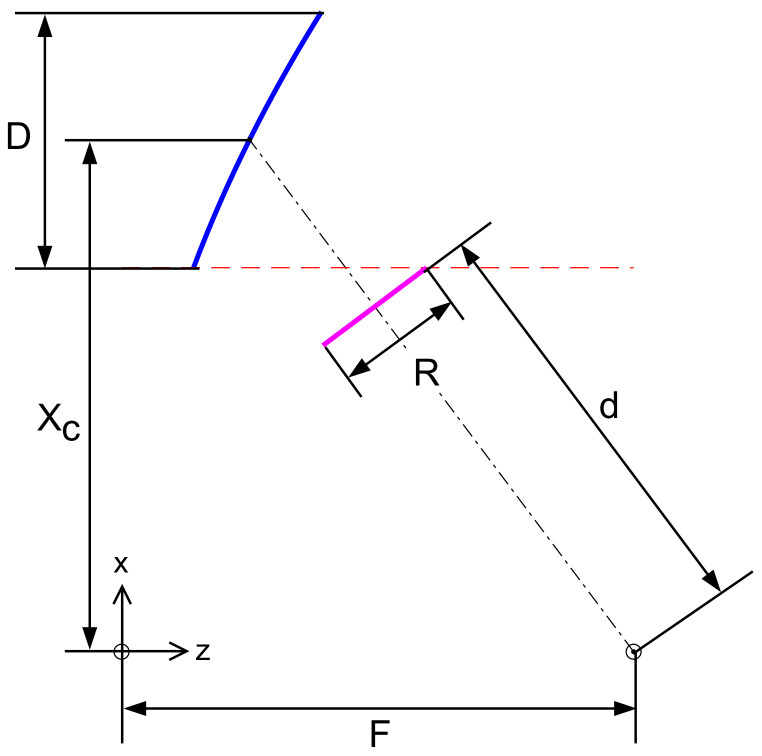
Schematic representation of the antenna system and design parameters.

**Figure 2 sensors-22-02586-f002:**
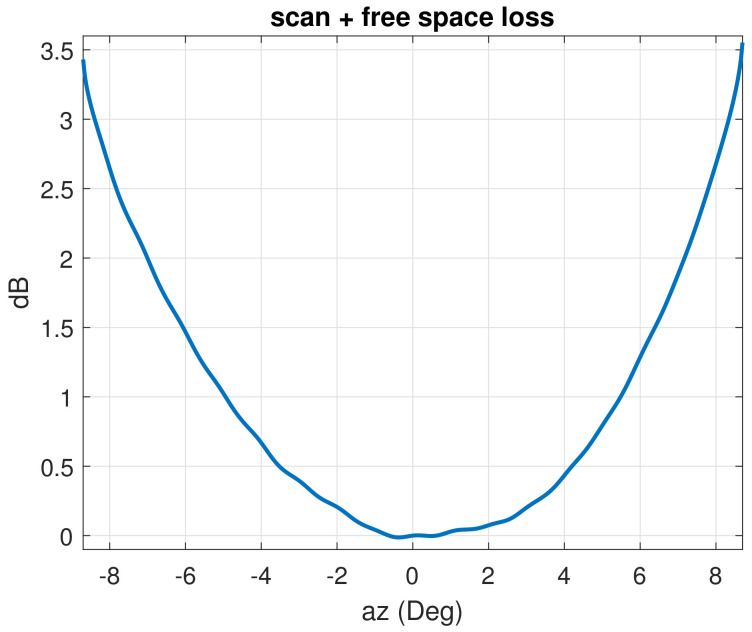
Scan losses plus propagation losses derived from antenna design and computed FSL as a function of azimuth in the plane of zero elevation.

**Figure 3 sensors-22-02586-f003:**
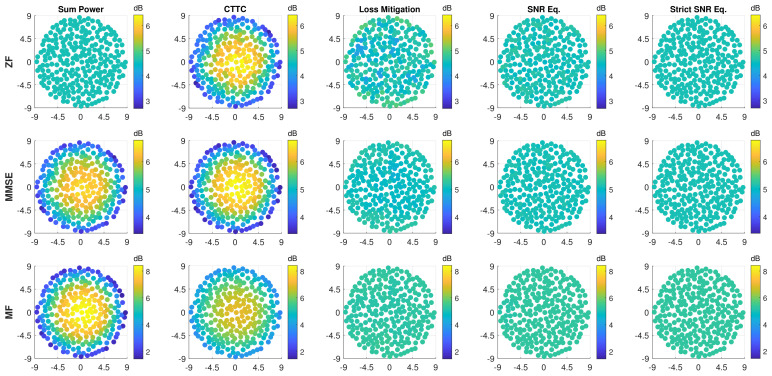
SNR (dB) per user versus position in azimuth and elevation in degrees for all discussed precoding and normalization methods. The rows span the precoding/beamforming techniques (ZF, MMSE, and MF), while the columns represent the normalization approaches (Sum Power, CTTC, Loss Mitigation, SNR Equalization, and Strict SNR Equalization). K=255.

**Figure 4 sensors-22-02586-f004:**
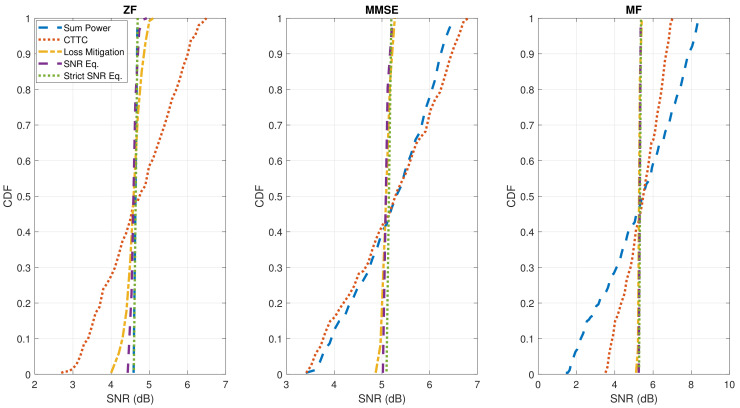
CDF of the obtained SNR (dB) per user for all discussed precoding and normalization methods. From left to right, the results for the different precoding methods (ZF, MMSE, and MF) are plotted. K=255.

**Figure 5 sensors-22-02586-f005:**
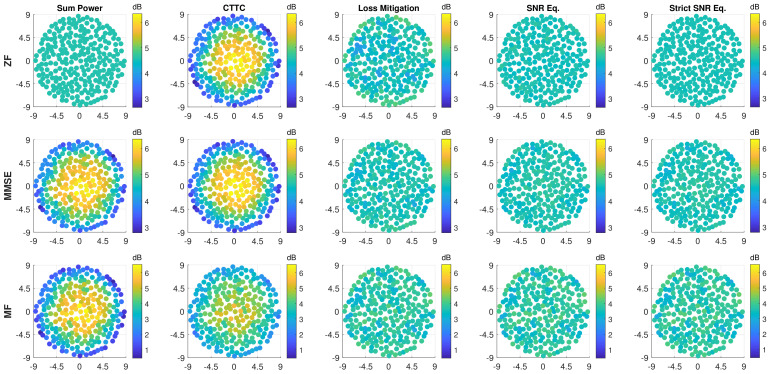
SNIR (dB) per user versus position in azimuth and elevation in degrees for all discussed precoding and normalization methods. The rows span the precoding/beamforming techniques (ZF, MMSE, and MF), while the columns represent the normalization approaches (Sum Power, CTTC, Loss Mitigation, SNR Equalization, and Strict SNR Equalization). K=255.

**Figure 6 sensors-22-02586-f006:**
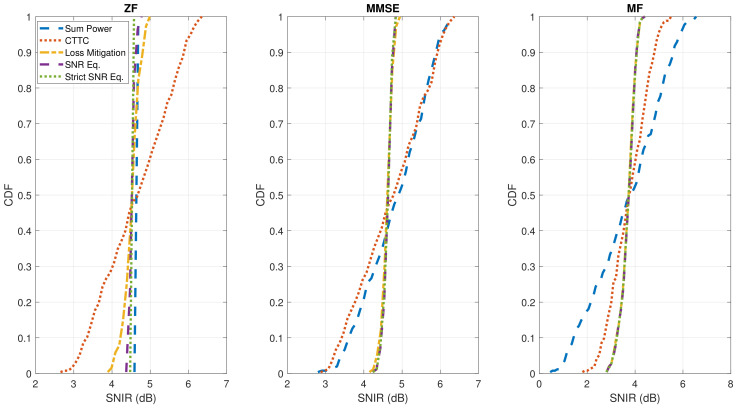
CDF of the obtained SNIR (dB) per user for all discussed precoding and normalization methods. From left to right, the results for the different precoding methods (ZF, MMSE, and MF) are plotted. K=255.

**Figure 7 sensors-22-02586-f007:**
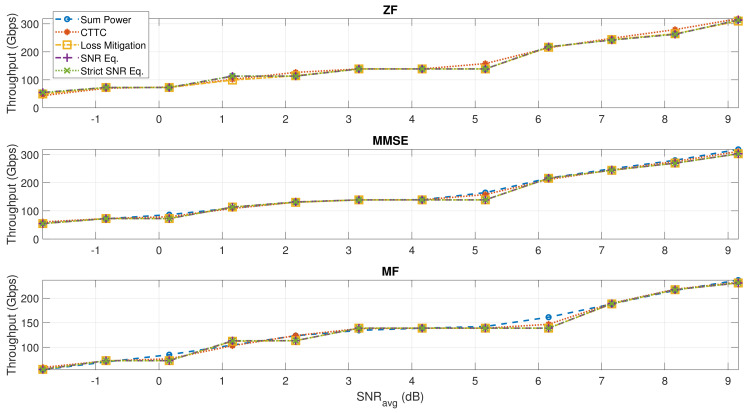
Total throughput at various $SNR_{avg}$ for all the different precoding and normalization techniques. K=255.

**Figure 8 sensors-22-02586-f008:**
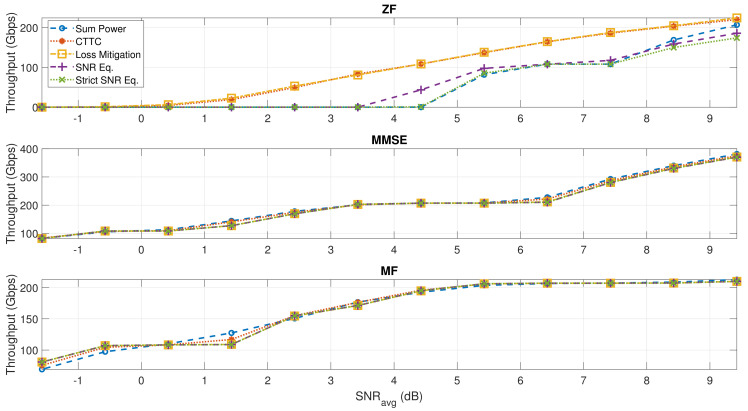
Total throughput at various $SNR_{avg}$ for all the different precoding and normalization techniques. K=380.

**Table 1 sensors-22-02586-t001:** Optimized design parameters.

Symbol	Parameter	Value
$f_{0}$	Frequency	17.7 GHz
*D*	Reflector aperture	2 m
*F*	Focal Length	4 m
$X_{c}$	Offset height	4 m
δ	Feed spacing	2.24 λ
*T*	Feed taper	−2 dB at 12 degrees
*N*	Number of feeds	511
*R*	Array aperture	900 mm
MG	Margin array aperture	57 mm (1.5 δ)
*d*	Defocused distance	3.37 m
*M*	Magnification	≈1.5

**Table 2 sensors-22-02586-t002:** CTTC steps.

Step 1:	$\tilde{U}=\frac{1}{\sqrt{K}}W\mathrm{diag}\left\{\frac{1}{p_{1}^{user}},\ldots ,\frac{1}{p_{K}^{user}}\right\}$.
Step 2:	$U=\frac{1}{\sqrt{N}}\mathrm{diag}\left\{\frac{1}{p_{1}^{feed}},\ldots ,\frac{1}{p_{N}^{feed}}\right\}\tilde{U}$.

**Table 3 sensors-22-02586-t003:** Loss Mitigation steps.

Step 1:	$\tilde{U}=\frac{1}{\sqrt{K}}W\mathrm{diag}\left\{\frac{f(\rho_{1})}{p_{1}^{user}},\ldots ,\frac{f(\rho_{K})}{p_{K}^{user}}\right\}$.
Step 2:	$U=\frac{1}{\sqrt{N}}\mathrm{diag}\left\{\frac{1}{p_{1}^{feed}},\ldots ,\frac{1}{p_{N}^{feed}}\right\}\tilde{U}$.

**Table 4 sensors-22-02586-t004:** SNR Equalization steps.

Step 1:	$\tilde{U}=W\mathrm{diag}\left\{\frac{1}{s_{1}},\ldots ,\frac{1}{s_{K}}\right\}$.
Step 2:	$U=\frac{1}{\sqrt{N}}\mathrm{diag}\left\{\frac{1}{p_{1}^{feed}},\ldots ,\frac{1}{p_{N}^{feed}}\right\}\tilde{U}$.

**Table 5 sensors-22-02586-t005:** Strict SNR Equalization steps.

Step 1:	$\tilde{U}=\frac{1}{\sqrt{N}}\mathrm{diag}\left\{\frac{1}{p_{1}^{feed}},\ldots ,\frac{1}{p_{N}^{feed}}\right\}$**W**.
Step 2:	$\bar{U}=\tilde{U}\mathrm{diag}\left\{\frac{1}{{\tilde{s}}_{1}},\ldots ,\frac{1}{{\tilde{s}}_{K}}\right\}$.
Step 3:	$U=\frac{\bar{U}}{\sqrt{\mathrm{trace}(\bar{U}{\bar{U}}^{H})}}$.

**Table 6 sensors-22-02586-t006:** System parameters for simulations.

Symbol	Parameter	Value
$N_{trials}$	Monte Carlo trials	5
$f_{0}$	Frequency	17.7 GHz
$B_{w}$	Bandwidth	500 MHz
*P*	Total RF power	3 kW
G/T	User terminal gain over receiver noise	17 dB/K
$D_{R}$	User terminal antenna diameter	0.75 m
$a_{eff}$	User terminal efficiency	75%
$R_{E}$	Earth radius	6378 km
lat	Satellite latitude	0
lon	Satellite longitude	13
alt	Satellite altitude	35,786 km
*K*	Number of simultaneous active users	255–380
$SNR_{ref}$	Reference SNR	3–5 dB

**Table 7 sensors-22-02586-t007:** Minimum and maximum SNR dynamic ranges over all the considered noise scenarios. K=255.

		Sum Power	CTTC	Loss Mitigation	SNR Eq.	Strict SNR Eq.
ZF	min	0.0 dB	3.5 dB	1.0 dB	0.3 dB	0.0 dB
	max	0.0 dB	3.5 dB	1.0 dB	0.3 dB	0.0 dB
MMSE	min	2.5 dB	3.4 dB	0.3 dB	0.0 dB	0.0 dB
	max	6.0 dB	3.5 dB	0.5 dB	0.2 dB	0.0 dB
MF	min	7.0 dB	3.5 dB	0.3 dB	0.0 dB	0.0 dB
	max	7.0 dB	3.5 dB	0.3 dB	0.0 dB	0.0 dB

**Table 8 sensors-22-02586-t008:** Minimum and maximum SNIR dynamic ranges over all the considered noise scenarios. K=255.

		Sum Power	CTTC	Loss Mitigation	SNR Eq.	Strict SNR Eq.
ZF	min	0.0 dB	3.4 dB	1.0 dB	0.3 dB	0.0 dB
	max	0.0 dB	3.5 dB	1.2 dB	0.3 dB	0.2 dB
MMSE	min	3.0 dB	3.3 dB	0.4 dB	0.3 dB	0.3 dB
	max	5.8 dB	3.4 dB	1.3 dB	1.1 dB	1.1 dB
MF	min	5.0 dB	3.2 dB	0.4 dB	0.3 dB	0.3 dB
	max	6.7 dB	3.4 dB	2.3 dB	2.2 dB	2.2 dB

**Table 9 sensors-22-02586-t009:** Difference of the power per feed from the reference uniform power (P/N) for all the considered noise scenarios. Minimum and maximum values refer to the feed with the lowest and highest power, respectively.

		Sum Power	CTTC	Loss Mitigation	SNR Eq.	Strict SNR Eq.
ZF	min	−1.4 dB	0.0 dB	0.0 dB	0.0 dB	0.0 dB
	max	1.4 dB	0.0 dB	0.0 dB	0.0 dB	0.0 dB
MMSE	min	−1.3 dB	0.0 dB	0.0 dB	0.0 dB	−0.2 dB
	max	1.3 dB	0.0 dB	0.0 dB	0.0 dB	0.2 dB
MF	min	−0.7 dB	0.0 dB	0.0 dB	0.0 dB	−0.2 dB
	max	0.4 dB	0.0 dB	0.0 dB	0.0 dB	0.1 dB

**Table 10 sensors-22-02586-t010:** Minimum and maximum SNR dynamic ranges over all the considered noise scenarios. K=380.

		Sum Power	CTTC	Loss Mitigation	SNR Eq.	Strict SNR Eq.
MMSE	min	3.1 dB	3.4 dB	0.3 dB	0.1 dB	0.0 dB
	max	5.6 dB	3.6 dB	1.4 dB	0.2 dB	0.0 dB
MF	min	7.0 dB	3.5 dB	0.3 dB	0.0 dB	0.0 dB
	max	7.0 dB	3.5 dB	0.3 dB	0.0 dB	0.0 dB

**Table 11 sensors-22-02586-t011:** Minimum and maximum SNIR dynamic ranges over all the considered noise scenarios. K=380.

		Sum Power	CTTC	Loss Mitigation	SNR Eq.	Strict SNR Eq.
MMSE	min	5.4 dB	3.5 dB	1.2 dB	1.2 dB	1.2 dB
	max	6.5 dB	6.7 dB	5.2 dB	4.7 dB	4.6 dB
MF	min	6.2 dB	3.5 dB	1.3 dB	1.3 dB	1.3 dB
	max	7.1 dB	6.2 dB	5.4 dB	5.5 dB	5.4 dB

## Data Availability

Not applicable.

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
