# Peer review of "Closed-Form Power Normalization Methods for a Satellite MIMO System"

_sensors, 2022, doi:10.3390/s22072586_

Round 1
Reviewer 1 Report
In the introductory section, the contributions of this work should be clearly presented, in order to facilitate the readers.
In section 2 system models is presented. The problem is mathematically formulated, with accurate and well written expressions. However, authors should consider citing some relevant literature in each equation used (not in the cases that equations are easily understandable).
In section 3, authors could possibly embed some algorithm tables for better understanding.
In section 4, the authors claim that 5 MC repeats are sufficient for such problems. This should be clearly elaborated, as this is a bit confusing.
• A key comment on this manuscript is that authors could have acted in the direction of proposing a scheme solving the presented limitations and not witnessing them.
Expand acronyms the first time they are shown both in abstract and in the body of the manuscript (i.e. MIMO, DVB-S2X, ZF, MMSE, MF etc.)
Reference SNR of 5 dB should be included in Table 2.
Enrich the reference list with relevant and recent papers, such as ET Michailidis, N Nomikos, P Trakadas, AG Kanatas, "Three-dimensional modeling of mmWave doubly massive MIMO aerial fading channels", IEEE Transactions on Vehicular Technology 69 (2), pp. 1190-1202, 2019.
Author Response
In the introductory section, the contributions of this work should be clearly presented, in order to facilitate the readers.
- Thank you for the suggestion, which will certainly help the reader grasp quickly the main contributions of our work. These are summarized as recommended at the end of the introduction in lines 116 to 123.
In section 2 system models is presented. The problem is mathematically formulated, with accurate and well written expressions. However, authors should consider citing some relevant literature in each equation used (not in the cases that equations are easily understandable).
- We agree with this recommendation and references are now included where relevant, specifically in Equations (3,4,6,8,13,18)
In section 3, authors could possibly embed some algorithm tables for better understanding.
- Thank you for the valuable suggestion. Tables 3,4,5,6 have been added describing the implemented algorithms.
In section 4, the authors claim that 5 MC repeats are sufficient for such problems. This should be clearly elaborated, as this is a bit confusing.
- We agree this may seem a bit low and probably not understood without further details. The claim that 5 iterations were sufficient was supported by reference to [28] in the original version of the manuscript. We have expanded that part and also included numerical results to demonstrate the claim remains valid with our assumptions. This can be found right after Table 7.
A key comment on this manuscript is that authors could have acted in the direction of proposing a scheme solving the presented limitations and not witnessing them.
- We kindly disagree with this statement in that we believe our work not only witnesses limitations but proposes power normalisation schemes towards mitigating it. As explained in the introduction, one way of mitigating antenna performance degradation across the service area is to shape the antenna pattern. This is proposed in the patent [18]. However, the performance is a combination of the antenna gain and the power amplification. So effectively, the scan losses and propagation losses can be mitigated by adjusting the antenna system, the power distribution at payload level or a combination of both. Our paper is meant to demonstrate the benefits of the second approach.
This aspect is better clarified in the introduction. We also mention in the conclusion that a combination of antenna shaping and power distribution control can be considered as future work.
Expand acronyms the first time they are shown both in abstract and in the body of the manuscript (i.e. MIMO, DVB-S2X, ZF, MMSE, MF etc.)
- We implemented the recommendation. Thank you for spotting this.
Reference SNR of 5 dB should be included in Table 2.
- Thank you for the suggestion. We have added the Number of simultaneous active users (255 and 380) and reference SNR (5 dB for 255 users and 3 dB for 380 users)
Enrich the reference list with relevant and recent papers, such as ET Michailidis, N Nomikos, P Trakadas, AG Kanatas, "Three-dimensional modeling of mmWave doubly massive MIMO aerial fading channels", IEEE Transactions on Vehicular Technology 69 (2), pp. 1190-1202, 2019.
- The following references have been added:
- Michailidis, E.T.; Nomikos, N.; Trakadas, P.; Kanatas, A.G. Three-Dimensional Modeling of mmWave Doubly Massive MIMO
Aerial Fading Channels. IEEE Transactions on Vehicular Technology 2020, 69, 1190–1202. doi:10.1109/TVT.2019.2956460. - Artiga, X.; Vázquez, M.A. Effects of channel phase in multibeam multicast satellite precoding systems. Advances in Communications Satellite Systems. Proceedings of the 37th International Communications Satellite Systems Conference (ICSSC-2019), 2019,
1–12. doi:10.1049/cp.2019.1271 - Taricco, G.; Ginesi, A. Precoding for Flexible High Throughput Satellites: Hot-Spot Scenario. IEEE Transactions on Broadcasting
2019, 65, 65–72. doi:10.1109/TBC.2018.2847438. - Cooley, M. Phased Array Fed Reflector (PAFR) antenna architectures for space-based sensors. 2015 IEEE Aerospace Conference,
2015, pp. 1–11. doi:10.1109/AERO.2015.7118963.
- Michailidis, E.T.; Nomikos, N.; Trakadas, P.; Kanatas, A.G. Three-Dimensional Modeling of mmWave Doubly Massive MIMO
Reviewer 2 Report
My comments are in the file i attachment.

Author Response
- You should cite more recent works as related works. The main contribution should be more highlighted to distinguish it from the recent existing works.
- Thank you for these important suggestions. The main contributions of the work are now highlighted at the end of the introduction in lines 116 to 123 and we added the following references:
- Michailidis, E.T.; Nomikos, N.; Trakadas, P.; Kanatas, A.G. Three-Dimensional Modeling of mmWave Doubly Massive MIMO
Aerial Fading Channels. IEEE Transactions on Vehicular Technology 2020, 69, 1190–1202. doi:10.1109/TVT.2019.2956460. - Artiga, X.; Vázquez, M.A. Effects of channel phase in multibeam multicast satellite precoding systems. Advances in Communications Satellite Systems. Proceedings of the 37th International Communications Satellite Systems Conference (ICSSC-2019), 2019,
1–12. doi:10.1049/cp.2019.1271 - Taricco, G.; Ginesi, A. Precoding for Flexible High Throughput Satellites: Hot-Spot Scenario. IEEE Transactions on Broadcasting
2019, 65, 65–72. doi:10.1109/TBC.2018.2847438. - Cooley, M. Phased Array Fed Reflector (PAFR) antenna architectures for space-based sensors. 2015 IEEE Aerospace Conference,
2015, pp. 1–11. doi:10.1109/AERO.2015.7118963.
- Michailidis, E.T.; Nomikos, N.; Trakadas, P.; Kanatas, A.G. Three-Dimensional Modeling of mmWave Doubly Massive MIMO
- It is preferable to describe the steps followed to obtain the presented results by an algorithm (pseudo code).
- We agree this can help to understand the process. We have added in Table 2 the algorithm steps to reproduce the results and Tables 3,4,5,6 the description of the normalization techniques
- It is preferable to study the performance of the proposed model under other environmental conditions such as non-line-of-sight communication and under other sources of perturbation, such as rain fading.
- Our paper is focusing on perturbations that are deterministic. Our proposed solution is an alternative to antenna reflector shaping, which has potential to reduce the cost of the antenna system with a more generic design. The perturbations mentioned by the reviewer are independent of the antenna system. Thus techniques developed for phased arrays could be applicable and are considered beyond the scope of this work.
- Figures 5 and 6 are not well commented on and interpreted. You should justify the difference in CDF for ZF, MMSE, and MF. Also, you should highlight the performance improvement that you have achieved by your proposed model.
- Thank you for noticing this, we have better commented these figures, discussing differences in precoding techniques and the achieved improvements, in section 4.2.1.
- From the obtained results, we can see that the throughput at various SNR has not been widely enhanced. However, there are few noise scenarios where the proposed normalization techniques perform slightly worse than CTTC. While in the majority of the SNRavg values the throughput results of the proposed techniques compared to CTTC are practically the same. So, you should mention the advantages of the proposed techniques and more justify how the proposed techniques can improve the throughput as compared to the existing techniques in the literature.
- We agree this can help to make the throughput results clearer and avoid confusion, we have stated the advantages in section 4.2.2.
- The Performance Analysis can be more enhanced by comparisons between your work and other more recent works in the form of numerical or plot presentations.
- To the best of our knowledge, the proposed approach has not been considered yet for AFR systems. As mentioned in the introduction, these systems are being considered for future software defined generic payloads and these investigations are considered a timely development. If the reviewer is aware of a recent work that is considered key for a benchmark in this specific field of research and can provide the reference, we would be pleased to include such a comparison.
- It is preferable to mention if the proposed TCDTF can be generalized for other frequency ranges and other communication systems.
- Thank you for this suggestion; in the discussion section we have improved the considerations about other communication systems and frequencies.
- The discussion section should be more developed by interpreting the obtained results in more detail. Furthermore, it is preferable to interpret the challenges introduced by the deployment of the proposed model. Also, the authors should focus on the Common challenging issues and growing research directions, by studying the balance between obtained accuracy, complexity, and energy efficiency.
- We agree these are important considerations, we have added key aspects for interpreting results and discussion on complexity and achievable benefits.
- You should add more recent references, especially for mathematical expressions, several facts, and some statistics.
- Thank you for this valuable suggestion. In many equations we are now citing related literature, specifically Equations (3,4,6,8,13,18), added these references to support facts and better explained statistical issues, like the Monte Carlo iterations right after Table 7.
- Lavrador, P.M.; Cunha, T.R.; Cabral, P.M.; Pedro, J. The Linearity-Effciency Compromise. IEEE Microwave Magazine 2010,
11, 44–58. doi:10.1109/MMM.2010.937100. - Maral, G.; Bousquet, M. Satellite Communications Systems, 4th ed.; Wiley, 2002
- Proakis, J.; Salehi, M. Digital Communications; McGraw-Hill International Edition, McGraw-Hill, 2008
- Brennan, D.G. Linear Diversity Combining Techniques. Proceedings of the IRE 1959, 47, 1075–1102. doi:10.1109/JRPROC.1959.287136.
- Lavrador, P.M.; Cunha, T.R.; Cabral, P.M.; Pedro, J. The Linearity-Effciency Compromise. IEEE Microwave Magazine 2010,
- The writing quality is excellent but it can be improved by avoiding typos and editing errors. There are some missed or added ‘s’ and ‘the’. Also, many lines are not numbered
- Thank you for having noted that. We have fixed line numbering and important typos.